# Effects of Substituents on the Photophysical/Photobiological Properties of Mono-Substituted Corroles

**DOI:** 10.3390/molecules28031385

**Published:** 2023-02-01

**Authors:** Vitória Barbosa de Souza, Vinícius N. da Rocha, Paulo Cesar Piquini, Otávio Augusto Chaves, Bernardo A. Iglesias

**Affiliations:** 1Bioinorganic and Porphyrinoids Materials Laboratory, Department of Chemistry, Federal University of Santa Maria, Santa Maria 97105-900, RS, Brazil; 2Department of Physics, Federal University of Santa Maria, Santa Maria 97105-900, RS, Brazil; 3CQC-IMS, Department of Chemistry, University of Coimbra, Rua Larga, 3004-535 Coimbra, Portugal

**Keywords:** corroles, *trans*-A_2_B-corroles, photophysics, photobiology

## Abstract

The *trans*-A_2_B-corrole series was prepared starting with 5-(pentafluorophenyl)dipyrromethene, which was then reacted with respective aryl-substituted aldehyde by Gryko synthesis. It was further characterized by HRMS and electrochemical methods. In addition, we investigated experimental photophysical properties (absorption, emission by steady-state and time-resolved fluorescence) in several solvents and TDDFT calculations, aggregation, photostability and reactive oxygen species generation (ROS), which are relevant when selecting photosensitizers used in photodynamic therapy and many other photo-applications. In addition, we also evaluated the biomolecule-binding properties with CT-DNA and HSA by spectroscopy, viscometry and molecular docking calculations assays.

## 1. Introduction

Tetrapyrrole macrocycles such as corroles are known as photosensitizers for use in photobiological processes and have recently drawn attention for their great capacity to generate reactive oxygen species (ROS), such as singlet oxygen, low cytotoxicity, desirable photostability and advantageous use of light at long absorption wavelengths up to near-infrared, which facilitates the application of these derivatives in photodynamic processes [1,2,3].

Corroles have a corrin-like skeleton with a direct bond between two pyrrole rings, as well as an 18-electron porphyrin-like π system, i.e., a hybrid structure between the two tetrapyrrole macrocycles. With recent synthetic improvements, the corrole moiety has attracted significant attention as a new photosensitizer [4,5]. These corroles have been extensively investigated by spectroscopic techniques (absorption/emission), theoretical molecular orbitals calculations, as well as by electrochemical studies. Understanding the general photophysics of corrole derivatives is also necessary for their application in photobiological processes such as photodynamic therapy (PDT) and antimicrobial photodynamic therapy (aPDT), as photophysical parameters such as excited state lifetime, intersystem crossing (ISC) rates and different quantum yields of the sensitizers have specific effects on the functionality of corroles, including their ability to generate ROS [6,7,8]. The presence of specific substituents in the *meso*-aryl positions of corrole can directly affect these parameters, meaning these compounds have different photophysical properties [9,10].

Interactive studies into Human Serum Albumin (HSA), the main carrier protein in the human bloodstream, are important as it is used in preliminary laboratory evaluations of pharmacokinetic parameters (offering interesting information before clinical evaluations) due to its ability to transport endogenous and exogenous compounds [11,12,13,14,15]. In the case of DNA interaction assays, the possible formation of a corrole-DNA adducts through intercalation or by major/minor groove interactions may induce structural changes in DNA and possible cell destruction by apoptosis or necrosis [16,17,18,19,20].

In this manuscript, four corrole derivatives **1**–**4** (Figure 1), which have different substituents at the *meso*-aryl position, were used, and their photophysical properties were investigated in combination with theoretical approaches (TDDFT). In addition, photobiological parameters and the interaction with biomacromolecules (DNA and HSA) were previously investigated by considering the effect of different types of substituents on the corrole macrocycle. The corrole structural change from phenyl (**1**), naphthyl (**2**), 4-(hydroxy)phenyl (**3**) or 4-(thiomethyl)phenyl (**4**) moiety in the *meso* position of the *trans*-C_6_F_5_ corrole was also evaluated in terms of its impact on the binding affinity with biomolecules.

## 2. Results

### 2.1. Corroles

Corroles **1**–**4** were prepared according to Gryko’s synthesis [21,22,23,24] and used as molecules to evaluate their photophysical/photobiological processes and interaction with biomolecules (CT-DNA and HSA). The HRMS-ESI(+) mass spectra of derivatives are presented in the Appendix A (Appendix A).

### 2.2. Photophysical Properties

The absorption spectra in the UV-Vis region of corroles **1**–**4** in several solvents (DCM, ACN, MeOH and DMSO) are shown in Figure 2 and the data referring to molar absorptivity (ε) and wavelengths of the main transitions (λ_abs_) are listed in Table 1.

In general, all corroles in both solvents showed electronic transitions already predicted for this type of derivative; in this case, the Soret band and two Q bands (Figure 2). For both corroles in ACN and DMSO solutions, the Soret band splitting and a difference in intensities of Q-bands were noted when compared to the DCM or MeOH solution. This fact has been reported in the literature by several authors and can be attributed to the presence of possible tautomeric species in a solution, occurring by the possibility of pyrrole nitrogen deprotonation or hydrogen bond interactions between pentafluorophenyl-corroles and the solvent [25,26]. On this occasion, different tautomeric types of the studied corroles are predicted in polar solvents such as ACN and DMSO, thus it should be expected that a possible thermodynamic equilibrium in the presence of both tautomeric species would be reached. Polar solvents can interact with the molecules by solvation and stabilize them, mainly through possible intermolecular interactions such as hydrogen bonds or dipole–dipole forces. This case is no longer observed in MeOH solution, as it is protic and acidic enough to keep the H atoms in the N-pyrrole macrocycle ring.

The steady-state emission fluorescence of corroles **1**–**4** in both solvents (λ_exc_ at Soret band and 600–800 nm range) and the photophysical data are reported in Table 1 and fluorescence emission spectra of corroles are shown in Figure 3. The emission quantum yield values (φ_F_) were calculated according to the reference tetra(phenyl)porphyrin standard in DMF solution (TPP; φ_F_ = 11.0%). In general, the values of the emission quantum yield of the studied corrole derivatives agree with the predicted structure (Table 1). More notable differences are observed in the DMSO solution, where the compounds in general showed the highest φ_F_ values, probably due to a greater stabilization of the excited states by the more polar solvent. The differences in the fluorescence data come from the presence of the substituents attached at the *meso*-10-aryl position of the corrole ring. In the MeOH solution, the change in the spectrum profile is also noticeable, with only one transition in the excited state, a fact that can be attributed to interactions by hydrogen bond interactions by the protic solvent (Figure 3). Data referring to HOMO-LUMO theoretical calculations by TDDFT analysis of these corroles will help a better interpretation of the presented results to be made.

With regard to the lifetime (**τ_f_**) values of the corroles **1**–**4**, no major changes were observed according to the polarity of the solvent, with lifetimes between the 3.0 and 5.0 ns range and non-radiative (*k*_nr_) rates higher than the radiative (*k*_r_) ones. The fluorescence decay plots of the compounds are shown in the Appendix A (Appendix A).

### 2.3. TDDFT Analysis

Table 2 summarizes the theoretical results for the absorption wavelengths, in nm, and oscillator strengths, ƒ, of the main peaks of the Soret and Q-bands in DCM for all the studied corroles. The natural transition orbitals relating to the lowest energy electronic excitation in the DCM solvent are also shown in Table 2. These data were obtained at the ground state equilibrium geometry of each compound. The theoretical optical absorption spectra and TDDFT data of two tautomeric states of corroles in DMSO are given in the Appendix A (Appendix A and Appendix A). Differences in the absorption spectra due to the implicit solvent environment are seen only for corrole **1** (Ph moiety), with the spectrum for DMSO showing longer absorption wavelengths for the Q bands and shorter wavelengths for the Soret band, compared to the spectrum in DCM. For compounds **2** to **4**, the calculated absorption spectra show practically the same profile in both solvents. Previous studies on corrole-like systems show the same behavior for these two solvents [5]. The theoretical absorption wavelengths for the Soret and Q bands are blue-shifted in DMSO as well as in DCM, compared to the experimental results. The reason for the deviations is related to the fact that the energies of the excited states are calculated at the ground state geometry.

To better understand the threshold for the optical transitions predicted by TDDFT calculations, the natural transition orbitals (NTOs) are shown in Table 2. These orbitals are associated with the lowest energy Q-band absorption peak for the DCM solvent. These NTOs for the DMSO solvent are very similar to those for the DCM solvent. For all studied cases, the NTOs associated with the S_0_ to S_1_ excitation show electronic transitions involving π-like orbitals distributed on the macrocyclic ring. The π→π* absorption peaks in this specific energy range indicate that these compounds can be good candidates for photodynamic therapy. An analysis of the excited states shows that there are two triplet states, T_1_ and T_2_, lower in energy than the first excited state, S_1_, that can be involved in intersystem crossing processes for all the studied corroles, as show in Appendix A. Furthermore, these two triplet states have energy gaps relative to the ground state that are larger than 0.98 eV, satisfying the energetic requirement to the generation of ^1^O_2_ species [27].

### 2.4. Aggregation, Stability in Solution and Photostability Assays

In the aggregation experiments, we observed a very low tendency of aggregation of the studied corroles **1**–**4** in both solvents, including in the DMSO (5%)/Tris-HCl pH 7.4 mixture buffered solution (for biological applications). A linear increase was observed in the absorption spectra as a function of the concentration variation from 2.0 to 30 µM. Both observations indicate that aggregation is not present in all assays. The UV-Vis spectra of corroles **1**–**4** in DCM, ACN, MeOH, DMSO and DMSO (5%)/Tris-HCl pH 7.4 buffered mixture solutions are presented in the Appendix A (Appendix A).

Regarding the stability of derivatives **1**–**4** in the solution, they were monitored by UV-Vis spectroscopy for a period of seven days (see Appendix A—Appendix A). The experiments were only conducted in DMSO solution and DMSO (5%)/Tris-HCl pH 7.4 buffered mixture due to their use in biological medium. In this way, we can conclude that these compounds are stable and can be used safely in this period of time.

Anticipating photodynamic applications, corroles **1**–**4** were submitted to photostability tests under white-light LED irradiation (fluence rate of 50 mW cm^−2^ and a total light dosage of 90 J cm^−2^) for a period of 20 min. Both corroles were tested in DMSO solution and DMSO (5%)/Tris-HCl pH 7.4 mixture. It is possible to note from the graphs in Figure 4 that in DMSO solution, the studied corroles are more susceptible to photodegradation processes than in the DMSO (5%)/Tris-HCl pH 7.4 media (Figure 4). This fact can be explained by the fact that in the absence of an aqueous medium, the derivatives can generate more ROS, thus increasing the photodegradation power of the derivatives. Another fact that may be related to this behavior is the fact that these reactive oxygen species have shorter lifetimes in water [28], thus favoring the integrity of the corroles.

### 2.5. Redox Analysis

The electrochemical profile by cyclic voltammetry of corroles **1**–**4** shows redox processes between −2.00 V and +2.00 V versus SHE in dry DCM solution at a scan rate of 100 mV s^−1^, using TBAPF_6_ as the supporting electrolyte. All CVs of the studied corroles are presented in the Appendix A (Appendix A) and the assignments of the redox peaks are presented in the Table 3. In the positive region, the oxidation processes of studied corroles exhibit three redox peak potentials for the scan between 0.0 V and +2.00 V (Table 3). The oxidation peaks of corroles are assigned to the formation of the mono, second and third mono-electronic oxidation peaks, generating π-cation radical, cation and di-cations, respectively [4,5]. Only in the case of corrole **4**, which contains a SCH_3_ substituent, can the first oxidation process be attributed to an oxidation of the S atom to a sulfoxide species (S=O). In the negative region and the cathodic region, only one reduction process was observed in both cases between the 0.0 V and −1.50 V potential range. The reduction process can be attributed to the π-anion radical species for corroles (Table 3).

### 2.6. Photobiological Assays

For singlet oxygen (^1^O_2_) generation, corroles **1**–**4** are suitable for phototherapeutic application (Table 4). All DPBF photooxidation UV-Vis spectra are compiled in the Appendix A (Appendix A). Compared to the standard *meso*-tri(phenyl)corrole (TPhCor) in DMSO solution, the presence of substituents in the *meso*-aryl position can interfere in ^1^O_2_ production, probably by intersystem crossing processes, mainly due to the presence of chemical groups attached in the corrole structure.

The capacity of corroles **1**–**4** to generate superoxide species (O_2_^•−^) was investigated in DMSO solution. For this application, solutions of corroles containing NBT and the reducing agent NADH were irradiated with a white-light LED source (fluence rate of 50 mW cm^−2^ and a total light dosage of 90 J cm^−2^) in aerobic conditions, at a period of 20 min. The reaction of NBT with superoxide radical species produced diformazan, which can be monitored following the absorption band of this product (Appendix A, Appendix A). The superoxide generation constant (*k*_SO_) by the NBT reduction assays is shown in Table 4. These results indicate that corroles **1**–**4**, after white-light irradiation conditions, can possibly form O_2_^•−^ species. The generation of superoxide is also dependent on the substituent inserted in corrole, since electron donating groups favor the formation of these species in a solution (Table 4).

Finally, the partition coefficients (log *P*_OW_) were measured for each corrole derivative and the values found for the neutral corroles containing different substituents are in accordance with the literature [29], with corroles **1** and **2** showing a more hydrophobic character when compared to the hydroxy-phenyl **3** and thio(methyl)phenyl **4** derivatives. This fact is attributed to the presence of more polar groups in the periphery of corroles, leaving compounds **3** and **4** with a more hydrophilic character (Table 4).

## 3. Biomolecule-Binding Assays

### 3.1. Binding Properties with DNA by UV-Vis Analysis

In order to evaluate the interaction between DNA and studied corroles **2**–**4**, UV-Vis absorption analysis was carried out. The UV-Vis spectra for the corroles in the absence and presence of successive additions of CT-DNA concentrations are shown in Figure 5, using compound **4** as an example, and the DNA-binding properties are listed in Table 5. All UV-Vis spectra of corroles **2** and **3** are presented in the Appendix A (Appendix A).

In this analysis, the successive additions of CT-DNA to corrole compound solutions caused hypochromic effects at the Soret and Q bands without a red or blue shift, indicating that the studied corroles can be interacted with CT-DNA, probably via secondary interactions by major or minor grooves around the biomacromolecule. Although neutral derivatives lack the potential for cationic–anionic electrostatic binding with DNA phosphate units, the corroles under study demonstrated an ability to interact with nucleic acids, corroborating previous studies that reported the activity of non-charged tetrapyrrolic macrocycles [13].

In this way, the binding constant (*K*_b_) values are in the 10^5^ M^−1^ range (Table 5), indicating that both corroles interact strongly with CT-DNA and the presence of substituents can interfere in the binding affinity of CT-DNA. Thermodynamic analysis via Gibbs free energy ΔG° values (Table 5) indicated that all corroles interacted spontaneously with CT-DNA, thus reinforcing the results observed by *K*_b_ values. In the next section, the possibilities of interaction between corroles in terms of intercalation or via grooves are investigated.

### 3.2. Competitive Binding Assays with DNA by Steady-State Fluorescence Emission

The steady-state fluorescence emission spectra involving the competition assays for the binding between corroles and DNA:dyes adducts are presented, using corrole **4** as an example (Figure 6). The fluorescence Stern–Volmer (*K*_SV_), bimolecular rate quenching (*k*_q_), binding (*K*_b_) and ΔG° parameters for DNA:dye:corroles are listed in Table 5.

As an example, Figure 6a depicts the fluorescence emission spectra for AO bound to CT-DNA (fluorescence emission at 534 nm when excited at 480 nm) in the absence and presence of corrole **4**. When corroles were added to CT-DNA pre-treated with acridine orange dye, corrole **4** induced a decrease in the fluorescence intensity of CT-AO:DNA adduct, indicating a displacement of AO from DNA, which can be assigned as a viable competition between AO and corroles for CT-DNA strands (specifically to the region rich in adenine and thymine). The Stern–Volmer (*K*_SV_), bimolecular quenching rate (*k*_q_) and binding (*K*_b_) constant values in the presence of each derivative are summarized in Table 5.

To evaluate whether the possible interaction of substituted corroles occurs via the groove site, the minor groove binder DAPI dye was used for the steady-state fluorescence emission quenching assays (Figure 6b). In independent experiments, it was possible to observe significant fluorescence quenching of DAPI:DNA upon successive additions of corrole **4**. Comparing both *K*_SV_ and *K*_b_ values for competitive binding assays into AO:DNA and DAPI:DNA adducts, it can be inferred that there is a significant variation in the fluorescence quenching constants, mainly in the presence of the studied corroles (Table 5). Overall, the *K*_SV_ and *K*_b_ data variation can be attributed to a preference for corrole interaction by the minor groove and not only by an intercalation phenomenon, agreeing with the molecular docking calculations (described in the next section) and the literature [13,20].

Moreover, the bimolecular quenching rate constant (*k*_q_~10^12^ M^−1^ s^−1^) for the corroles, being higher than the diffusion rate constant according to Smoluchowski–Stokes–Einstein theory [30], indicates a ground state association between the corroles and nucleobases in the DNA strands (probably static fluorescence quenching mechanism). Additionally, the fluorescence emission spectra for the other corroles into AO:DNA and DAPI:DNA solutions are presented in the Appendix A (Appendix A).

### 3.3. Viscosity Measurements with DNA

It is known that viscosity assays are sensitive to the change in the length of the DNA double helix. In the absence of other structural techniques, viscosity measurements are considered an important method to determine intercalation or non-intercalation binding of compounds to DNA nucleobases. The results of the viscosity measurements are shown in Appendix A in the Appendix A. The DNA viscosity remains almost unchanged upon the addition of corroles **1**–**4**, with an increase in the ratio [corrole]/[CT-DNA]. These results indicate that the studied corroles do not intercalate between the DNA bases, corroborating the fluorescence emission DAPI experiments (probably binds to minor grooves). This is an expected result, since these corrole derivatives have a steric volume that is not conducive to promoting some intercalation phenomenon.

### 3.4. Binding Properties with HSA by UV-Vis and Emission Analysis

In addition to the UV-Vis analysis for DNA, the interaction of the studied corroles with HSA was also tested and the absorption spectra are listed in the Appendix A (Appendix A). In general, the values found for interactions such as *K*_b_ and ΔG° are lower than for the interaction with CT-DNA, as can be seen in Table 6.

For HSA albumin interactions, the fluorescence emission was monitored by gradually increasing the corrole derivative concentration in DMSO (5%)/Tris-HCl pH 7.4 mixture buffered solution; for example, compound **3** is presented in Figure 7. The HSA fluorescence emission quenching spectra of all corroles are shown in the Appendix A (Appendix A).

To examine the nature of the fluorescence quenching mechanism induced by these corrole derivatives, the Stern–Volmer equation was applied. The results showed a good linear relationship and the Stern–Volmer quenching constants (*K*_SV_ and *k*_q_) (Table 6). The values observed for the bimolecular quenching rate constant (*k*_q_~10^12^ M^−1^ s^−1^) are three orders of magnitude larger than the diffusion rate constants according to Smoluchowski–Stokes–Einstein [30], corroborating that the nature of quenching is a possible static mechanism. The association constant (*K*_a_), binding constant (*K*_b_), and the number of binding sites (*n*) were obtained using the modified Stern–Volmer and double-logarithm equations [13,20]. The *K*_a_ and *K*_b_ constants correlated well with the values of the Stern–Volmer quenching constants (*K*_SV_) and were obtained from the emission assays for the interactions of derivatives with HSA. In addition, since the *n* values are variable (1.23 to 1.95), the derivatives probably bind to HSA in different sites; hence, a SF and time-resolved analysis was made to check this possibility (see next section).

### 3.5. Synchronous Fluorescence (SF) Analysis

Comparing the steady-state fluorescence technique with the synchronous fluorescence (SF), the latter has been considered a complementary and more sensitive approach to detect possible perturbations in the microenvironment around the two main fluorophores of albumin (tyrosine and tryptophan residues) after drug binding [31]. Appendix A in the Appendix A show the SF spectra for HSA without and upon successive additions of corroles **1**–**4** at Δλ 15 nm and 60 nm for Tyr and Trp residues, respectively. In general, for both Δλ there is a significant decrease in the fluorescence signal upon additions of corrole compounds; however, it did not induce any blue- or red-shift, agreeing with the steady-state fluorescence data (Section 3.4), which indicated that the binding of corroles does not induce any significant perturbation on the microenvironment around the fluorophores.

### 3.6. Time-Resolved Fluorescence with HSA and Corroles **1**–**4**

In order to identify the main fluorescence quenching mechanism (static or dynamic), time-resolved fluorescence decays were obtained for HSA without and in the presence of corroles **1**–**4** in DMSO(5%)/Tris-HCl pH 7.4 buffered mixture solution and lifetime plots; values are presented in the Appendix A (Appendix A and Appendix A). There was a decrease in the τ_f_ value of HSA upon corrole addition, indicating that at high derivative concentration a combined static and dynamic fluorescence quenching mechanism is feasible for the interaction HSA:corroles.

### 3.7. Molecular Docking Analysis with DNA and HSA

Molecular docking calculations are interesting tools to provide the molecular aspects of the interaction between small compounds and biomacromolecules [32,33]. Thus, to determine the binding modes and binding sites of the studied corroles **2**–**4** with both DNA and HSA, a blind docking was performed in the major and minor grooves of DNA and in the three main binding sites of albumin. Molecular docking analysis between DNA and HSA with corrole **1** has been previously described by Acunha and co-workers [13].

Firstly, Table 7 and Figure 8 depict the docking score value (dimensionless) and docking pose for DNA:corroles. The GOLD 2020.2 software ranked the ten best docking poses for each system (biomacromolecule-ligand) and the best result was treated as shown in Figure 8. Since the docking score value of each pose is considered as the negative value of the sum of energy terms from the mechanical-molecular type component, which includes intramolecular tensions in the ligand and intermolecular interactions in the biomacromolecule-ligand association, the more positive the score value (dimensionless), the better the interaction. According to the computational results, corroles **2**–**4** can be accommodated mainly in the minor groove of DNA (highest docking score value, e.g., 61.19 and 40.37 for DNA:**2** in the minor and major grooves, respectively—Table 7), which is in agreement with experimental competitive binding assays.

In the case of HSA there are three main binding pockets for endogenous and exogenous compounds: subdomain IIA (site I), located in a hydrophobic binding pocket where Trp-214 residues can be found, subdomain IIIA (site II), also located in a hydrophobic binding pocket, and subdomain IB (site III), located on the surface of the albumin (Figure 9). Table 7 shows the docking score value (dimensionless) for the three main binding sites. Since the highest docking score value for the corrole derivatives was obtained for site III, molecular docking results suggested that the external pocket (subdomain IB) is the main binding region for compounds **3** and **4**, while site I was the main binding site for derivative **2**. Although all corroles under study possess a high steric volume that might impact the preference for binding in the external region of albumin, it is probable that corrole **2** interacts mainly in the internal pocket of albumin due to the high lipophilicity of the compound. It has been previously described that fluorinated-phenyl- and pyrenyl-corroles bind mainly into the site I of albumin [13], indicating that the presence of hetero-atoms in the phenyl moieties of the corroles under study changes the polarity of the corrole and the capacity of interaction with an external pocket.

Molecular docking results suggested van der Waals and hydrogen bonding as the main binding forces responsible for both interactions DNA:corroles and HSA:corroles (Appendix A in the Appendix A, and Figure 8 and Figure 9. In silico calculations suggested that the replacement of hydroxy-phenyl (corrole **3**) to thio(methyl)phenyl (corrole **4**) moieties did not change the binding conformation to albumim.

## 4. Materials and Methods

### 4.1. General

All chemical reagents were of analytical grade and purchased from Sigma-Aldrich^®^ and Oakwood Chemical^®^ (Estill, SC, USA) without any further purification. The calf-thymus acid desoxyribonucleic (CT-DNA) and human serum albumin (HSA) was lyophilized powder and fatty acid-free (Sigma-Aldrich^®^, St. Louis, MO, USA; purity ≥99%). The concentration of the stock solutions of biomolecules was confirmed by UV-Vis analysis through the Beer–Lambert equation with a molar absorptivity (ε) value of 6600 M^−1^ cm^−1^ for CT-DNA at 260 nm (per nucleic acid) and 35,700 M^−1^ cm^−1^ for HSA at 280 nm in Tris-HCl buffer (pH 7.4) solution; the water used in all experiments was milliQ grade. Compounds were analyzed using a high-resolution mass spectrometer with electrospray ionization (HRMS-ESI) in the positive mode using a micrOTOF-QII mass spectrometer (Bruker Daltonics, Billerica, MA, USA). Mass spectra were recorded for each sample in the methanolic solution (concentration of 500 ppb) with a flow of 5.0 μL/min and capillarity of 6000 V.

### 4.2. Photophysical Measurements

Absorption UV-Vis spectra were obtained using the Shimadzu UV-2600 spectrophotometer (1.0 cm optical path length) and measured in the 250–800 nm region for studied corroles in several solvents such as acetonitrile (ACN), dichloromethane (DCM), methanol (MeOH), dimethyl sulfoxide (DMSO) and DMSO(5%)/Tris-HCl pH 7.4 mixture buffered solution, with fixed concentrations of 10 µM. For the determination of the steady-state fluorescence emission spectra, we employed a fluorimeter Horiba Fluoromax Plus, where the corroles were dissolved at a fixed concentration of 5.0 µM, in a 1.0 cm optical path length cuvette, excited at Soret transition band (slit 5.0; Em/Exc).

The fluorescence quantum yields (ϕ_f_) of the corrole derivatives were measured according to the literature, using tetra(phenyl)porphyrin (TPP) in DMF solution as the standard molecule [34]. Fluorescence lifetimes (τ_f_) were recorded using the Time Correlated Single Photon Counting (TCSPC) method with DeltaHub controller in conjunction with Horiba fluorimeter. Data were processed with DAS6 and Origin^®^ 8.5 software (Northampton, MA, USA) using mono-exponential fitting of raw data. NanoLED (Horiba) source (1.0 MHz, 441 nm excitation wavelength) was used as an excitation source. The instrumental response function (IRF) was collected using a Ludox^®^ dispersion (Sigma-Aldrich^®^, St. Louis, MO, USA). Radiative (*k*_r_) and non-radiative (*k*_nr_) values were calculated by equations according to the literature [20].

### 4.3. TDDFT Calculations

The electronic, structural, and optical properties of the studied compounds were determined using the Density Functional Theory (DFT) and its time-dependent extension (TDDFT). The ground state geometrical structures were optimized through conjugated gradient techniques. The wB97XD functional was used to represent the exchange and correlation potential [35], while the molecular orbitals were described by linear combinations of 6-31G(d,p) quality basis sets. We employed the polarizable continuum model (PCM) [36] to calculate the molecular properties either in dimethyl sulfoxide (DMSO) or dichloromethane (DCM) (ε = 46.826 and ε = 8.930, respectively). Natural Transitions Orbitals (NTOs) were calculated for the molecular orbitals involved in the lowest energy electronic transitions [37]. All calculations have been made using the Gaussian 09 quantum chemistry package [38].

### 4.4. Electrochemical Analysis

Cyclic voltammograms were recorded with a potenciostat/galvanostat AutoLab Eco Chemie PGSTAT 128N system at room temperature and under argon atmosphere in dry dichloromethane (DCM) solution. Electrochemical grade tetrabutylammonium hexafluorophosphate (TBAPF_6_, 0.1 M) was used as a supporting electrolyte. Employing a standard three-component system these CV experiments were carried out with: a glassy carbon working electrode; a platinum wire auxiliary electrode; and a platinum wire *pseudo*-reference electrode. To monitor the reference electrode, the Fc/Fc^+^ redox couple was used as an internal reference [39].

### 4.5. Photobiological Parameters

Aggregation by UV-Vis absorption analysis was conducted as a function of successive increase in corrole concentration (2.0 to 30 μM) in all solvents and changes in the λ_Soret_ in the 250–700 nm range were monitored according to the related literature [5]. The stability experiments in pure dimethyl sulfoxide (DMSO) and in DMSO(5%)/Tris-HCl pH 7.4 mixture buffered solution of studied corrole was also monitored by absorption UV–Vis measurements at several days (one to seven days). All experiments were performed in duplicate and independently.

The photostability assays in dimethyl sulfoxide (DMSO) and in DMSO(5%)/Tris-HCl pH 7.4 mixture buffered solution of related corrole derivatives was also monitored by absorption UV-Vis measurements at different exposure times (0 to 30 min) under the white-light LED array system (400 to 800 nm) at a fluence rate of 50 mW cm^−2^ and a total light dosage of 90 J cm^−2^. All experiments were performed in duplicate and independently, according to equations described in the literature [5].

Singlet oxygen (^1^O_2_) production was recorded according to typical 1,3-diphenylisobenzofuran (DPBF) photooxidation assays; the maximum volume of 1.0 mL which contained 100 μM DPBF in DMSO was mixed with 0.5 mL (50 μM) of corroles **1**–**4**. The flask was then filled with 2.0 mL of DMSO to a final volume of 3.5 mL. In order to measure singlet oxygen generation (φ_Δ_), absorption UV-Vis spectra of each solution were recorded at different exposure times (0 to 600 s, using red-light diode laser; Thera Laser DMC—São Paulo; potency of 100 mW) and φ_Δ_ were calculated according to the literature [4].

The superoxide radical (O_2_^•−^) species by NBT reduction assays was used and this approach was carried out at the same conditions stated in the literature, using Nitro Bule Tetrazolium (NBT) and NADH in DMSO solution [40]. Control experiments were performed in the absence of corroles and all derivatives were irradiated under aerobic conditions with a white-light LED source (fluence rate of 50 mW cm^−2^ and a total light dosage of 90 J cm^−2^) at a period of 20 min. The progress of the reaction was monitored by following the increase in the absorbance close to 560 nm. The superoxide generation constant (*k*_SO_) values can be obtained according to the literature cited above.

The partition coefficient (log *P*_OW_) of corroles **1**–**4** was determined using *n*-octanol (3.0 mL) and water (3.0 mL), according to the literature [20]. The corrole concentrations as well as their respective absorbances were determined by the UV-Vis absorption at 250 to 800 nm range, and the Soret band was chosen for monitoring and calculating the log *P*_OW_ values.

### 4.6. Biomacromolecule Interactive Studies

UV-Vis absorption analysis for each corrole without and in the presence of successive additions of CT-DNA or HSA solution were obtained at 298.15 K in DMSO(5%)/Tris-HCl pH 7.4 mixture buffered solution in the 250 to 700 nm range. The corrole concentration was fixed in 5.0 μM and CT-DNA or HSA was in the 0 to 100 μM range. The hyperchromicity (*H*%), red-shift (Δλ), binding constant (*K*_b_), and Gibb’s free-energy (ΔG°) values of the corroles **1**–**4** were calculated according to the literature, through Benesi-Hildebrand and free-energy equations [13,20]. An interactive DNA study using absorption analysis with corrole **1** has been previously described by Acunha and co-workers [13].

Competitive binding assays between CT-DNA:dyes and corroles by steady-state fluorescence emission analysis are recorded and corroles **2**–**4** in DMSO(5%)/Tris-HCl pH 7.4 mixture buffered solution (0 to 100 μM) were gradually added in a fixed concentration of acridine orange (AO; A-T rich intercalator; 10 μM ; λ_exc_= 490 nm, λ_em_= 500–800 nm) and 4′,6-diamidino-2-phenylindole (DAPI; minor groove binder; 10 μM; λ_exc_= 359 nm, λ_em_= 380–700 nm) and CT-DNA (10 μM) in DMSO(5%)/Tris-HCl pH 7.4 mixture buffered solution. The DNA:dye adducts were incubated for 3 min after corrole addition for each measurement. The Stern–Volmer quenching (*K*_SV_) and bimolecular quenching rate (*k*_q_) constants of corroles were calculated according to the DNA:dye fluorescence quenching using a plot of F_0_/F versus [corrole] and a ratio of *K*_SV_/τ_0_, where the τ_0_ denote the fluorescence lifetime of DNA:dye (AO = 2.20 ns and DAPI = 1.70 ns), respectively [13,20]. Binding (*K*_b_) constant and free-energy interaction (ΔG°) values are obtained by double-logarithm and Gibb’s equation according to the literature [20]. Competitive DNA:dye study by emission analysis with corrole **1** has been previously described by Acunha and co-workers [13].

Additionally, viscosity analysis was carried out using an Ostwald viscometer immersed in a water bath maintained at 298.15 K, according to the literature, with some modifications [41]. The CT-DNA concentration was kept constant in all compounds, while the corrole concentration was increased in DMSO (5%)/Tris-HCl pH 7.4 mixture buffered solution. The flow time was measured at least three times with a digital stop-watch (Casio^®^, Shibuya City, Tokyo, Japan) and the mean value was calculated. Data are presented as (η/η^0^)^1/3^ versus the ratio [corrole]/[CT-DNA], where η and η^0^ are the specific viscosity of CT-DNA in the presence and absence of the corroles **1**–**4**, respectively. The values of η and η^0^ were calculated by use of the expression (t − t_b_)/t_b_, where t is the observed flow time and t_b_ is the flow time of solution alone. The relative viscosity of the CT-DNA was calculated from η/η^0^.

For HSA-binding assays by steady-state fluorescence emission analysis to obtain quantitative parameters on the binding capacity of HSA:corroles **1**–**4**, the maximum fluorescence data after inner filter correction [31] were obtained via the Stern–Volmer (*K*_SV_) quenching constant, binding (*K*_b_) constant, and Gibbs’ free energy (ΔG°) values by Stern–Volmer and double-logarithmic equations, according to the literature [20]. All emission spectra were obtained by measurements at 298.15 K in DMSO(5%)/Tris-HCl pH 7.4 mixture buffered solution in the 300 to 500 nm range ([HSA] = 10 μM; [corroles] = 0 to 100 μM).

The synchronized fluorescence (SF) spectra were recorded for HSA (10 μM) without and in the presence of corroles **1**–**4** (concentration ranging from 0 to 100 μM) at room temperature (298.15 K). Spectra were recorded in the 240–320 nm range by setting Δλ = 15 nm (for tyrosine residue) and Δλ = 60 nm (for tryptophan residue).

Fluorescence lifetime decays (τ_f_) were recorded using Time Correlated Single Photon Counting (TCSPC) method with DeltaHub controller in conjunction with Horiba Jobin-Yvon Fluoromax Plus spectrofluorometer. Data were processed with DAS6 and Origin^®^ 8.5 software using mono-exponential fitting of raw data. NanoLED (Horiba) source (1.0 MHz, Pulse width < 1.2 ns at 284 nm excitation wavelength) was used as an excitation source. The HSA and corroles concentration were fixed in 10 μM each in DMSO (5%)/Tris-HCl pH 7.4 mixture buffered solution. Additionally, interactive binding properties with HSA with corrole **1** by spectroscopic methods has been previously described by Acunha and co-workers [13].

### 4.7. Molecular Docking Procedure with DNA and HSA

The crystallographic structure for DNA and HSA was obtained from the Protein Data Bank with access code 1BNA and 1N5U, respectively [39,40]. The chemical structure for corroles **2**, **3** and **4** was built and minimized in terms of energy by Density Functional Theory (DFT) calculations under B3LYP potential with basis set 6–31G*, available in the Spartan’18 software (Wavefunction, Inc., Irvine, CA, USA) [42,43,44]. The molecular docking calculations were performed with GOLD 2020.2 software (Cambridge Crystallographic Data Centre, Cambridge, CB2 1EZ, UK) [45].

Hydrogen atoms were added to the biomacromolecules following tautomeric states and ionization data inferred by GOLD 2020.2 software at pH 7.4. For DNA structure, the 10 Å radius around the center of mass of DT-20 was analyzed to explore the two main possible binding sites (major and minor grooves) [46]. On the other hand, for HSA structure 10 Å radius around the selected center of mass from the amino acid residue present in one of the three main binding pockets, more specifically Trp-214, Tyr-411, and Tyr-161 residues, for sites I, II, and III, respectively, was delimitated for molecular docking calculations. These amino acid residues were chosen according to the crystallographic structure of each site probe into HSA (warfarin, ibuprofen, and camptothecin) [47,48]. The number of genetic operations (crossing, migration, mutation) during the search procedure was set as 100,000. The software optimizes the geometry for hydrogen bonding by allowing the rotation of hydroxyl and amino groups contained in the biomacromolecules. The side chain rotamers have been defined according to the availability of the library. For all biomacromolecules, ChemPLP was used as the scoring function due to the lowest Root Mean Square Deviation (RMSD) obtained in previous work for small organic compounds, including porphyrins [26,49,50,51]. The figures for the best docking pose were generated with the PyMOL Delano Scientific LLC software (Schrödinger, New York, NY, USA) [52].

## 5. Conclusions

In this study, we investigated and studied the photophysical/photobiological properties of substituted corroles containing different groups at *meso*-10-position (phenyl, naphthyl, 4-hydroxy-phenyl and 4-thio(methyl)phenyl) in some solvents. Photobiological parameters such as ROS generation and photostability were evaluated and it was found that these compounds are promising for use in photoinduced processes. Furthermore, the interactive properties of corroles **1**–**4** against biomacromolecules such as DNA and HSA were evaluated, and the corrole derivatives had a preference for interacting in the minor grooves of the DNA due to secondary forces, which are more evident in site III of the albumin.

## Figures and Tables

**Figure 1 molecules-28-01385-f001:**
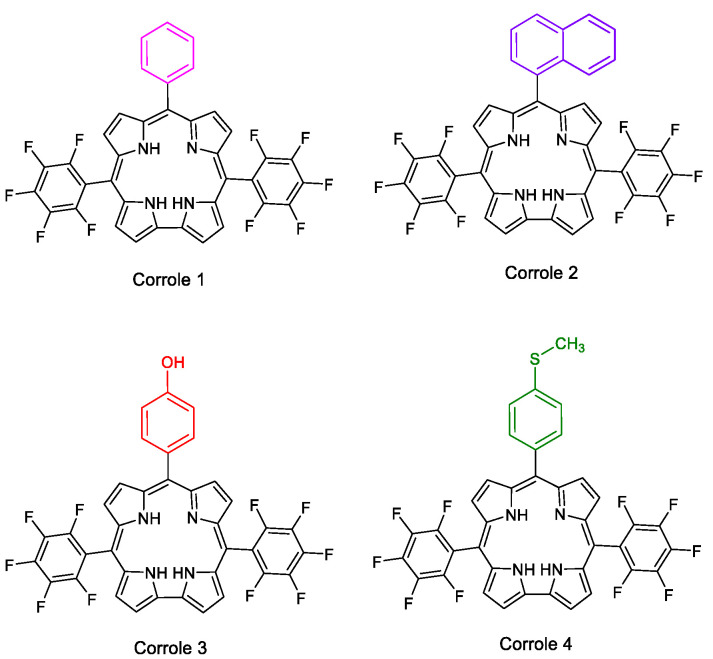
Chemical representative structure of mono-substituted corroles **1**–**4**.

**Figure 2 molecules-28-01385-f002:**
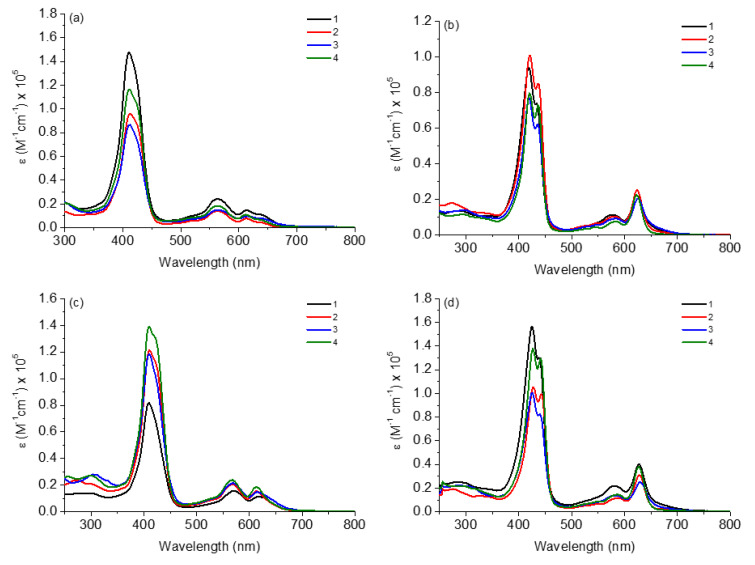
Absorption UV-Vis spectra of corroles **1**–**4** in (**a**) DCM, (**b**) ACN, (**c**) MeOH and (**d**) DMSO, at concentration of 10 μM.

**Figure 3 molecules-28-01385-f003:**
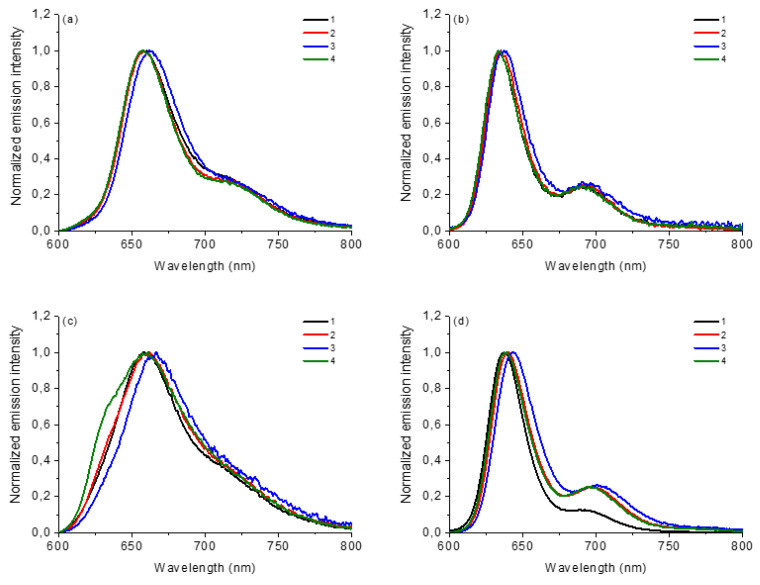
Normalized steady-state fluorescence emission spectra of corroles **1**–**4** in (**a**) DCM, (**b**) ACN, (**c**) MeOH and (**d**) DMSO, at concentration of 5.0 μM (λ_exc_ at Soret transition band).

**Figure 4 molecules-28-01385-f004:**
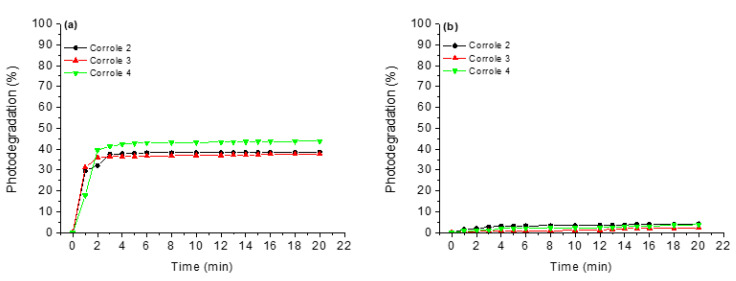
Photostability assays under white-light LED irradiation (fluence rate of 50 mW cm^−2^ and a total light dosage of 90 J cm^−2^) for a period of 20 min, where (**a**) DMSO and (**b**) DMSO(5%)/Tris-HCl pH 7.4 mixture buffered solution.

**Figure 5 molecules-28-01385-f005:**
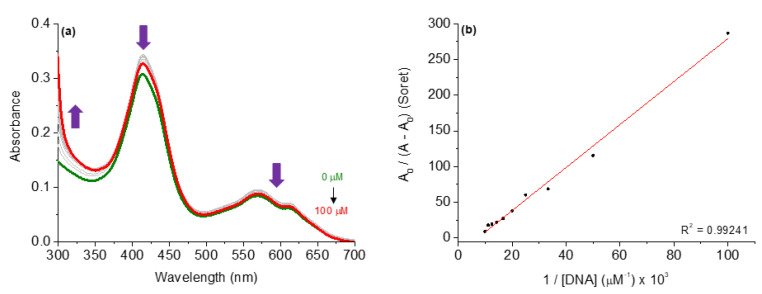
(**a**) UV-Vis spectra of the corrole **4** upon successive additions of CT-DNA concentrations (0 to 100 µM) in DMSO(5%)/Tris-HCl pH 7.4 mixture buffered solution. (**b**) Benesi-Hidelbrandt plots of A_0_/(A − A_0_) vs. 1/[CT-DNA].

**Figure 6 molecules-28-01385-f006:**
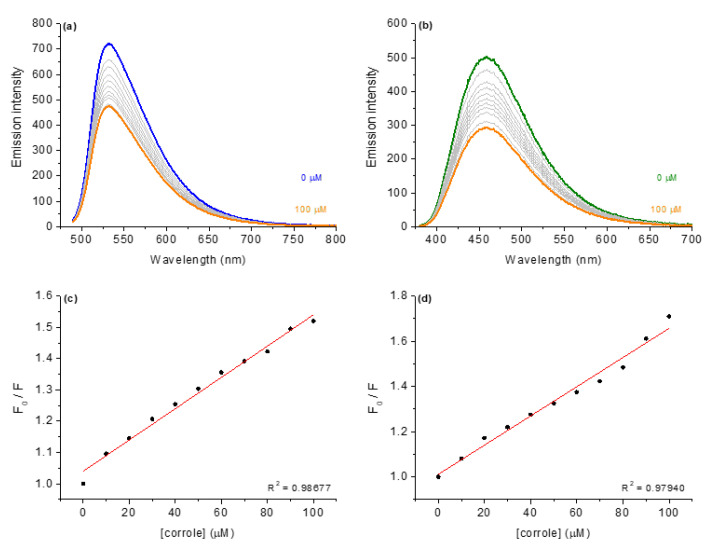
Steady-state fluorescence emission spectra for (**a**) AO:DNA and (**b**) DAPI:DNA without and in the presence of corrole **4**, in in DMSO(5%)/Tris-HCl pH 7.4 mixture buffered solution. Graphs (**c**,**d**) shows the plot F_0_/F versus [corrole]. [corrole] = 0–100 μM.

**Figure 7 molecules-28-01385-f007:**
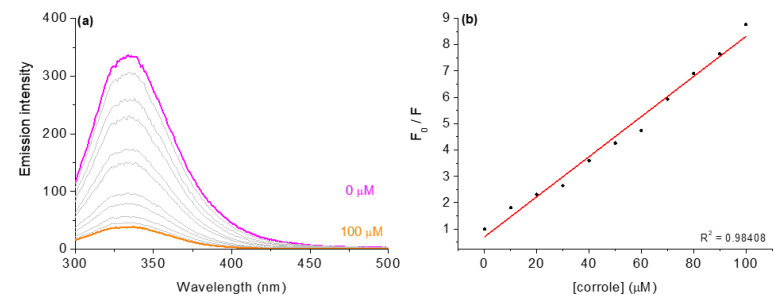
Steady-state fluorescence emission spectra for (**a**) HSA without and in the presence of corrole **3**, in DMSO(5%)/Tris-HCl pH 7.4 mixture buffered solution. Graph (**b**) shows the plot F_0_/F versus [corrole]. [corrole] = 0–100 μM.

**Figure 8 molecules-28-01385-f008:**
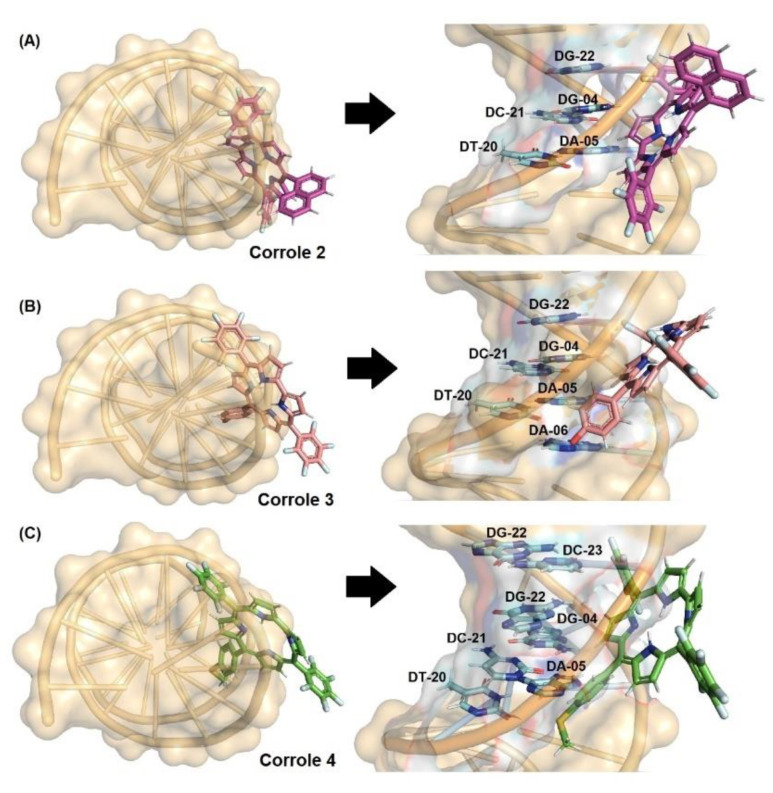
Best docking pose for (**A**) DNA:**2**, (**B**) DNA:**3**, and (**C**) DNA:**4** in the minor groove. Selected nitrogenated bases and corroles are in stick representation in cyan, pink, beige, green, and brown, respectively. Elements’ color: hydrogen: white; oxygen: red; nitrogen: dark blue, fluorine: light blue, and sulfur: yellow.

**Figure 9 molecules-28-01385-f009:**
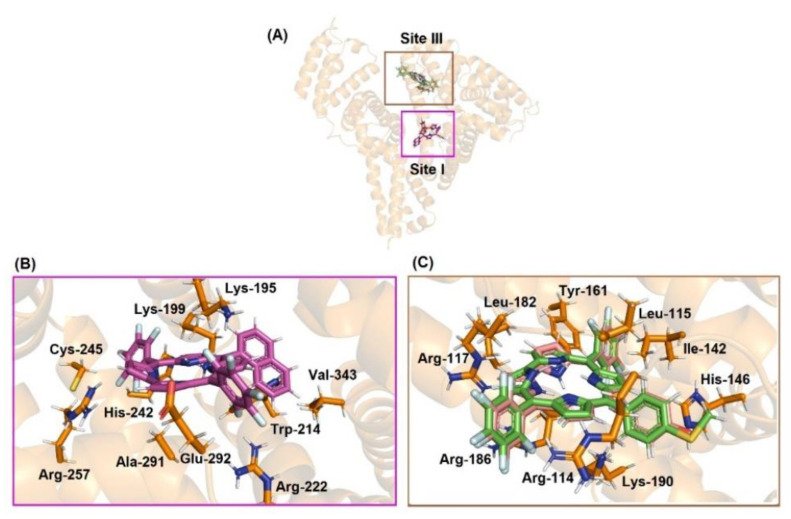
(**A**) Superposition of the best docking pose for HSA:corroles in the Sites I and III. Best docking pose for (**B**) HSA:**2** into site I and (**C**) HSA:**3/4** into site III. Selected amino acid residue and corroles are in stick representation in orange, pink, beige, green, and brown, respectively. Elements’ color: hydrogen: white; oxygen: red; nitrogen: dark blue, fluorine: light blue, and sulfur: yellow.

**Table 1 molecules-28-01385-t001:** Photophysical data of corroles **1**–**4**.

	DCM					
Corrole	λ_Abs_ (ε; M^−1^ cm^−1^ × 10^5^) ^a^	λ_Em_ (QY; %) ^b^	SS (nm/cm^−1^) ^c^	τ_f_ (ns) ^d^/Χ^2^	*k*_r_ (×10^8^ s^−1^) ^e^	*k*_nr_ (×10^8^ s^−1^) ^e^
**1**	411 (1.47), 563 (0.24), 614 (0.14)	659, 717 (15.0)	248/9155	3.66/1.05815	0.41	2.32
**2**	412 (0.96), 562 (0.14), 614 (0.07)	658, 719 (9.0)	246/9075	4.20/0.97026	0.21	2.16
**3**	411 (0.86), 563 (0.15), 615 (0.09)	662, 718 (8.0)	251/9225	3.70/1.16247	0.21	2.49
**4**	411 (1.16), 562 (0.18), 612 (0.11)	657, 719 (12.0)	246/9110	3.87/1.14992	0.31	2.27
	**ACN**					
**Corrole**	**λ_Abs_ (ε; M^−1^ cm^−1^ × 10^5^) ^a^**	**λ_Em_ (QY; %) ^b^**	**SS (nm/cm^−1^) ^c^**	**τ_f_ (ns)^d^/Χ^2^**	***k*_r_ (×10^8^ s^−1^) ^e^**	***k*_nr_ (×10^8^ s^−1^) ^e^**
**1**	419 (0.94), 577 (0.11), 625 (0.20)	635, 693 (11.0)	216/8120	4.04/1.05903	0.27	2.20
**2**	420 (1.01), 580 (0.10), 624 (0.25)	634, 692 (17.0)	214/8035	4.24/1.15462	0.40	1.96
**3**	419 (0.76), 583 (0.09), 627 (0.20)	637, 694 (8.0)	218/8165	3.68/1.13217	0.22	2.50
**4**	420 (0.79), 583 (0.07), 623 (0.22)	633, 692 (11.0)	213/8010	4.07/1.17485	0.27	2.18
	**MeOH**					
**Corrole**	**λ_Abs_ (ε; M^−1^ cm^−1^ × 10^5^) ^a^**	**λ_Em_ (QY; %) ^b^**	**SS (nm/cm^−1^) ^c^**	**τ_f_ (ns)^d^/Χ^2^**	***k*_r_ (×10^8^ s^−1^) ^e^**	***k*_nr_ (×10^8^ s^−1^) ^e^**
**1**	409 (0.82), 571 (0.15), 617 (0.11)	658 (11.0)	249/9250	3.77/1.13135	0.29	2.36
**2**	410 (1.21), 567 (0.20), 613 (0.18)	661 (6.0)	251/9260	3.78/1.01780	0.16	2.48
**3**	410 (1.17), 568 (0.21), 613 (0.15)	666 (22.0)	256/9375	3.56/1.07740	0.62	2.19
**4**	409 (1.39), 567 (0.24), 613 (0.18)	658 (10.0)	249/9250	3.66/1.05354	0.27	2.46
	**DMSO**					
**Corrole**	**λ_Abs_ (ε; M^−1^ cm^−1^ × 10^5^) ^a^**	**λ_Em_ (QY; %) ^b^**	**SS (nm/cm^−1^) ^c^**	**τ_f_ (ns) ^d^/Χ^2^**	***k*_r_ (×10^8^ s^−1^) ^e^**	***k*_nr_ (×10^8^ s^−1^) ^e^**
**1**	425 (1.56), 580 (0.22), 627 (0.40)	637, 692 (38.0)	212/7830	4.26/1.00523	0.89	1.45
**2**	427 (1.05), 587 (0.11), 628 (0.31)	640, 697 (26.0)	213/7795	4.20/1.10980	0.62	1.76
**3**	426 (1.00), 579 (0.14), 630 (0.25)	643, 701 (22.0)	217/7920	4.12/1.08084	0.53	1.89
**4**	427 (1.37), 585 (0.15), 627 (0.38)	639, 696 (27.0)	212/7770	4.35/1.09784	0.62	1.68

^a^ [ ] = 10 μM; ^b^ [ ] = 5.0 μM, using TPP as standard (QY = 11%, DMF solution); ^c^ SS = Stokes shifts = λ_em_ − λ_Soret_ (nm) = 1/λ_Soret_ − 1/λ_em_ (cm^−1^); ^d^ Excitation by NanoLED source at 441 nm; ^e^ According ref. [20].

**Table 2 molecules-28-01385-t002:** HOMO-LUMO plots for the lowest energy electronic transitions (S_0_→S_1_), transition energies in nm, and oscillator strengths (in parentheses), ƒ, for the main peak at the Soret band, and the two main peaks of the Q-band in DCM, for corroles **1**–**4**. The data for the Soret peak was calculated as the average for the two theoretically intense peaks (see Appendix A). This was done to allow a better comparison with the experimental results.

Corrole	HOMO Plot	LUMO Plot	Soret Band (*f*)	Q Bands (*f*)
**1**	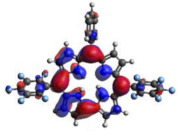	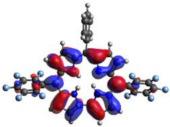	390.98 (1.1256)	**535.89 (0.1006);** **555.81 (0.2818)**
**2**	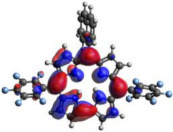	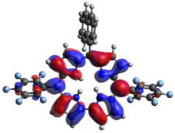	367.33 (1.3793)	**547.84 (0.1095);** **564.03 (0.2953)**
**3**	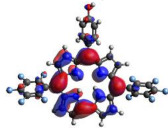	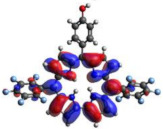	371.05 (1.3838)	**554.66 (0.0884); ** **575.63 (0.3254)**
**4**	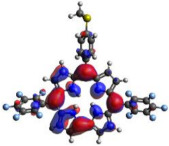	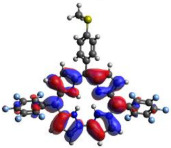	371.99 (1.4419)	553.48 (0.0835); 573.47 (0.3171)

**Table 3 molecules-28-01385-t003:** Redox potentials of corroles **1**–**4** in dry DCM solution (*E* versus SHE).

Corrole	*E_ox1_*	*E_ox2_*	*E_ox3_*	*E_red1_*	E_HOMO_ ^d^	E_LUMO_ ^e^	ΔE
**1 ***	+0.65 V ^a^	+0.78 V ^c^	+0.95 V ^c^	−1.29 V ^b^	−5.45	−3.51	1.94
**2**	+0.80 V ^a^	+1.32 V ^c^	-	−1.18 V ^b^	−5.60	−3.62	1.98
**3**	+0.60 V ^a^	+1.09 V ^c^	+1.93 V ^c^	−1.21 V ^b^	−5.40	−3.59	1.81
**4**	+0.21 V ^a^	+0.95 V ^c^	+1.62 V ^c^	−1.37 V ^b^	−5.01	−3.43	1.58

^a^ Anodic peak = *E*_pa_; ^b^ Cathodic peak = *E*_pc_; ^c^
*E*_1/2_ = *E*_pa_ + *E*_pc_/2; ^d^ E_HOMO_ = −[4.8 + *E*_ox1_ (versus SHE]eV; ^e^ E_LUMO_ = −[4.8 + *E*_red1_ (versus SHE]eV; * Ref. [5].

**Table 4 molecules-28-01385-t004:** Photobiological parameters of corroles **1**–**4**.

Corrole	*k*_po_ (M^−1^ s^−1^) ^a^	φ_Δ_ (%) ^b^	*k*_SO_ (min^−1^) ^c^	log *P_OW_* ^d^
**1 ***	0.322	33.0	0.641	+2.948
**2**	0.504	51.0	0.540	+3.542
**3**	0.444	45.0	0.913	+2.431
**4**	0.390	40.0	0.513	+2.135
**TPhCor ****	0.650	67.0	-----	-----

^a^ DPBF photooxidation constant in DMSO solution; ^b^ Singlet oxygen quantum yields in DMSO solution; ^c^ Superoxide formation constant in DMSO solution; ^d^ Water-octanol partition coefficients; * Ref. [5]; ** Ref. [5].

**Table 5 molecules-28-01385-t005:** DNA-binding properties of corroles **1**–**4** by UV-Vis and fluorescence emission analysis.

	UV-Vis Analysis				
Corrole	*H*(%) ^a^	Δλ (nm) ^b^	*K*_b_ (×10^5^; M^−1^) ^c^	ΔG° (kcal mol^−1^) ^d^	
**1 ***	6.10	0.0	1.49 ± 0.02	−7.05	
**2**	10.6	0.0	3.21 ± 0.07	−7.50	
**3**	3.80	0.0	1.75 ± 0.10	−7.15	
**4**	10.7	0.0	3.32 ± 0.03	−7.55	
	**AO:DNA by Emission**				
**Corrole**	** *Q* ** **(%) ^e^**	** *K* ** ** _SV_ ** **(×10^3^; M** ** ^−^ ** ** ^1^ ** **) ^f^**	** *k* ** ** _q_ ** **(×10^12^; M** ** ^−^ ** ** ^1^ ** **s** ** ^−^ ** ** ^1^ ** **) ^g^**	** *K* ** ** _b_ ** **(×10^3^; M** ** ^−^ ** ** ^1^ ** **) ^h^**	**Δ** **G° (kcal mol** ** ^−^ ** ** ^1^ ** **) ^d^**
**1 ***	25.0	3.43 ± 0.13	2.02	-----	-----
**2**	7.40	0.80 ± 0.01	0.47	1.16 ± 0.18	−4.18
**3**	33.3	4.27 ± 0.07	2.51	3.41 ± 0.12	−4.82
**4**	34.2	5.00 ± 0.10	2.94	2.71 ± 0.06	−4.68
	**DAPI:DNA by Emission**				
**Corrole**	** *Q* ** **(%) ^e^**	** *K* ** ** _SV_ ** **(×10^3^; M** ** ^−^ ** ** ^1^ ** **) ^f^**	** *k* ** ** _q_ ** **(×10^12^; M** ** ^−^ ** ** ^1^ ** **s** ** ^−^ ** ** ^1^ ** **) ^i^**	** *K* ** ** _b_ ** **(×10^3^; M** ** ^−^ ** ** ^1^ ** **) ^h^**	**Δ** **G° (kcal mol** ** ^−^ ** ** ^1^ ** **) ^d^**
**1 ***	37.4	5.81 ± 0.11	2.64	-----	-----
**2**	21.2	1.96 ± 0.02	0.89	3.90 ± 0.09	−4.90
**3**	33.8	3.43 ± 0.03	1.56	3.42 ± 0.13	−4.82
**4**	41.5	6.47 ± 0.02	2.94	3.29 ± 0.16	−4.80

^a^*H*(%) = (A_0_ − A)/A × 100%; ^b^ Red-shift; ^c^ Binding constant by Benesi-Hidelbrandt equation; ^d^ Determined by Gibbs free-energy equation; ^e^
*Q*(%) = (F_0_ − F)/F × 100%; ^f^ Determined by Stern–Volmer quenching constant; ^g^ Determined by *K*_SV_/τ_0_ ratio, where τ_0_ = 1.70 ns (AO:DNA); ^h^ Determined by double-logarithm equation; ^i^ Determined by *K*_SV_/τ_0_ ratio, where τ_0_ = 2.20 ns (DAPI:DNA).

**Table 6 molecules-28-01385-t006:** HSA-binding properties of corroles **1**–**4** by UV-Vis and fluorescence emission analysis.

	UV-Vis Analysis						
Corrole	*H*(%) ^a^	Δλ (nm) ^b^	*K*_b_ (×10^3^; M^−1^) ^c^	ΔG° (kcal mol^−1^) ^d^			
1	17.4	0.0	4.83 ± 0.04	−5.00			
2	7.25	0.0	4.76 ± 0.04	−5.00			
3	16.0	0.0	3.30 ± 0.07	−4.80			
4	29.0	0.0	16.4 ± 0.07	−5.75			
	**Emission Analysis**						
**Corrole**	** *Q* ** **(%) ^e^**	** *K* ** ** _SV_ ** **(×10^4^; M** ** ^−^ ** ** ^1^ ** **) ^f^**	** *k* ** ** _q_ ** **(×10^12^; M** ** ^−^ ** ** ^1^ ** **s** ** ^−^ ** ** ^1^ ** **) ^g^**	** *K* ** ** _a_ ** **(×10^4^; M** ** ^−^ ** ** ^1^ ** **) ^h^**	** *K* ** ** _b_ ** **(×10^4^; M** ** ^−^ ** ** ^1^ ** **) ^i^**	** *n^j^* **	**Δ** **G° (kcal mol** ** ^−^ ** ** ^1^ ** **) ^d^**
1 *	-----	1.41 ± 0.01	2.49	2.29 ± 0.13	-	-	−5.85
2	65.7	1.91 ± 0.05	3.37	1.00 ± 0.11	5.22 ± 0.19	1.23	−6.45
3	88.6	7.60 ± 0.18	13.4	8.75 ± 0.23	8.63 ± 0.42	1.95	−6.75
4	77.2	2.10 ± 0.06	3.70	9.13 ± 0.25	6.29 ± 0.39	1.45	−6.75

^a^*H*(%) = (A_0_ − A)/A × 100%; ^b^ Red-shift; ^c^ Binding constant by Benesi-Hidelbrandt equation; ^d^ Determined by Gibbs free-energy equation; ^e^
*Q*(%) = (F_0_ − F)/F × 100%; ^f^ Determined by Stern–Volmer quenching constant; ^g^ Determined by *K*_SV_/τ_0_ ratio, where τ_0_ = 5.67 ns (HSA); ^h^ Determined by modified Stern–Volmer equation; ^i^ Determined by double-logarithm equation, where *n* is the number of binding sites.

**Table 7 molecules-28-01385-t007:** Molecular docking score values (dimensionless) for the interaction between DNA/HSA and each corrole derivative under study into the corresponding main binding site.

	DNA		HSA		
Corrole	Minor Groove	Major Groove	Site I	Site II	Site III
**2**	61.19	40.37	83.03	30.30	74.09
**3**	63.75	39.87	53.84	30.90	72.20
**4**	65.60	43.86	69.31	30.88	76.72

## Data Availability

All analyzed data are contained in the main text of the article. Raw data are available from the authors upon request.

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
