# Peer review of "Effects of Substituents on the Photophysical/Photobiological Properties of Mono-Substituted Corroles"

_molecules, 2023, doi:10.3390/molecules28031385_

Round 1

Reviewer 1 Report

In this paper, the authors presented a comprehensive description of the new trans-A2B-corrole's, including both the synthesis and an in-depth description of the spectroscopic properties of the proposed compounds. The proposed work is an interesting article covering a wide spectrum of applied research methods, including experimental measurements and the use of computational chemistry methods. I recommend allowing the article to be published after making corrections covering its selected aspects.

1.       lack of literature links in the field of synthesis of the considered group of chemical compounds (line 64)

2.       The analysis of the spectroscopic properties of the examined compounds proposed by the authors is complete, taking into account the connection of both experimental and calculation data, therefore the extended hypotheses from experimental values should also be fully supported by appropriate calculations. Pointing to the occurrence of two tautomeric forms for ACN and DMSO solvents (Line 75) requires the attachment of relevant calculations for both tautomeric forms and to indicate for them the appropriate spectral values explaining the obtained experimental data. The authors pointed to the analysis of frontal orbits, however, it is largely limited to their visualization, and the analysis of their energy value and the corresponding Energy Gapp's for the solvents used would explain to the change in the location of the absorption maxima’s observed in the experiment.

3.       The proposed description of the docking methods does not in any way make it possible to repeat or verify the performed procedure, neither in the case of DNA and HSA, the coordinates of the grid box center are not given, the only known value is its radius of 10 A. Therefore, based on the initial crystallographic data for both biomolecules, it should be precisely specify the location of the centers of active sites analyzed in the simulation.

4.       I used many docking programs in which the value quantifying the result was measurable and expressed in energy units, which did not cause difficulties in their interpretation. If the applied GOLD program describes the obtained conformers with some indefinite value, its meaning should be clearly explained because the presented differences are difficult to unambiguously interpret. For example, Is the degree of fit to be understood on a scale of 0 to 100??

Author Response

In this paper, the authors presented a comprehensive description of the new trans-A2B-corrole's, including both the synthesis and an in-depth description of the spectroscopic properties of the proposed compounds. The proposed work is an interesting article covering a wide spectrum of applied research methods, including experimental measurements and the use of computational chemistry methods. I recommend allowing the article to be published after making corrections covering its selected aspects.

  1. lack of literature links in the field of synthesis of the considered group of chemical compounds(line 64)

Response: Thanks for the comment. Sorry for the mistake made and the references suggested by the reviewer have been duly added to the manuscript.

  1. The analysis of the spectroscopic properties of the examined compounds proposed by the authors is complete, taking into account the connection of both experimental and calculation data, therefore the extended hypotheses from experimental values should also be fully supported by appropriate calculations. Pointing to the occurrence of two tautomeric forms for ACN and DMSO solvents (Line 75) requires the attachment of relevant calculations for both tautomeric forms and to indicate for them the appropriate spectral values explaining the obtained experimental data. The authors pointed to the analysis of frontal orbits, however, it is largely limited to their visualization, and the analysis of their energy value and the corresponding Energy Gapp's for the solvents used would explain to the change in the location of the absorption maxima’s observed in the experiment.

Response: The theoretical calculations reveal that both tautomers show a split in the Soret band for the DMSO implicit solvent, suggesting that the split is not due to the presence of tautomers. We have performed calculations for DMSO and DCM, with this split in the Soret band appearing in both solvents. It should be emphasized that our theoretical calculations are done at 0K and using implicit solvent models.

  1. The proposed description of the docking methods does not in any way make it possible to repeat or verify the performed procedure, neither in the case of DNA and HSA, the coordinates of the grid box center are not given, the only known value is its radius of 10 A. Therefore, based on the initial crystallographic data for both biomolecules, it should be precisely specify the location of the centers of active sites analyzed in the simulation.

Response: The reviewer is totally right in his/her comment. Therefore, the information on the molecular docking procedure (section 4.7) for DNA and HSA was improved in the revised version of the manuscript.

  1. I used many docking programs in which the value quantifying the result was measurable and expressed in energy units, which did not cause difficulties in their interpretation. If the applied GOLD program describes the obtained conformers with some indefinite value, its meaning should be clearly explained because the presented differences are difficult to unambiguously interpret. For example, Is the degree of fit to be understood on a scale of 0 to 100??

Response: The GOLD 2020.2 software ranked the ten best docking poses for each system (biomacromolecule-ligand). Since the docking score value of each pose is considered as the negative value of the sum of energy terms from mechanical-molecular type component, which includes intramolecular tensions in the ligand and intermolecular interactions in the biomacromolecule-ligand association, more positive the score value (dimensionless) indicates better interaction (please, see https://www.ccdc.cam.ac.uk/solutions/csd-discovery/components/gold/). This information was added in the section 3.7 of the revised version of the manuscript.

Reviewer 2 Report

The authors report the photophysical properties of corrole derivatives with different aromatic substituents at the meso position. It was demonstrated that these aryl-substituted corroles were applicable to photodynamic therapy. In addition, other biochemical properties, such as biomolecule-binding properties, were also evaluated. The reviewer think that the contents are important in the field not only of corrole chemistry but also of photochemistry and biomedical applications. Therefore, this reviewer recommends publication of this manuscript in Molecules after addressing the following points:

1.     p. 2, section 2.1, l. 1: Please provide the appropriate reference(s) for Gryko's synthesis.

2.     p. 7: The authors mentioned "Compared to the standard meso-tri(phenyl)corrole (TPhCor) in DMSO solution, the presence of substituents ins the meso-aryl position can interfere in the 1O2 production, probably by intersystem crossing processes, mainly due to the presence of chemical groups attached in the corrole structure." Do you have any information about the mechanism that the meso substituent influences the rate of the intersystem crossing? I recommend TD-DFT calculations for their triplet states.

Author Response

The authors report the photophysical properties of corrole derivatives with different aromatic substituents at the meso position. It was demonstrated that these aryl-substituted corroles were applicable to photodynamic therapy. In addition, other biochemical properties, such as biomolecule-binding properties, were also evaluated. The reviewer think that the contents are important in the field not only of corrole chemistry but also of photochemistry and biomedical applications. Therefore, this reviewer recommends publication of this manuscript in Molecules after addressing the following points:

  1. p. 2, section 2.1, l. 1: Please provide the appropriate reference(s) for Gryko's synthesis.

Response: Thanks for the comment. As already answered to reviewer #1, the suggested references were duly added to the manuscript.

  1. p. 7: The authors mentioned "Compared to the standard meso-tri(phenyl)corrole (TPhCor) in DMSO solution, the presence of substituents ins the meso-aryl position can interfere in the 1O2 production, probably by intersystem crossing processes, mainly due to the presence of chemical groups attached in the corrole structure." Do you have any information about the mechanism that the meso substituent influences the rate of the intersystem crossing? I recommend TD-DFT calculations for their triplet states.

Response: The theoretical calculations show that the intersystem crossing can occur between the S1 state and the T1 or T2 triplet states, since these triplet states are the only ones below in energy in relation to the S1 state, as shown in Figs. S17-S18 in the Supplementary Material. Further, both T1 and T2 have an energy gap with the S0 ground state larger than 0.98 eV, satisfying the energy requirements to the generation of 1O2.
